



# An unconstrained formulation for complex solution phase minimization

Nicolas Riel [1,2], Boris J.P. Kaus [1,2,3], Albert de Montserrat [4], Evangelos Moulas [1,2,3], Eleanor C.R. Green [5], and Hugo Dominguez [1]

[1]Institute of Geosciences, Johannes Gutenberg-University, Mainz, Germany
[2]Terrestrial Magmatic Systems (TeMaS) research center, Johannes Gutenberg-University, Mainz, Germany
[3]Mainz Institute of Multiscale Modelling (M[3]ODEL), Johannes Gutenberg-University, Mainz, Germany
[4]Department of Earth Sciences, Institut für Geophysik, ETH Zürich, 8092 Zürich, Switzerland
[5]School of Geography, Earth and Atmospheric Sciences, The University of Melbourne, Victoria 3010, Australia

**Correspondence:** Nicolas Riel  (nriel@uni-mainz.de)

**Abstract.**

Prediction of mineral phase assemblages is essential to better understand the dynamics of the solid Earth, such as metamorphic processes, magmatism and the formation of mineral ore deposits. While recently developed thermodynamic databases allow the prediction of stable phase mineral assemblages for an increasing range of pressure, temperature and compositional spaces, the increasing complexity of these databases results in a significant increase of computational cost, hindering our ability to perform realistic models of reactive fluid/magma transport. Presently, prediction of stable phase equilibrium in complex systems is therefore largely limited by how efficiently single phase minimization can be performed, as more than 75 % of the total computational time is generally dedicated to individual solution phase minimization. This limitation becomes critical for non-ideal solution phase models that involve both a large number of chemical components, and mixing on a large number of sites, resulting in many inequality constraints of the form $0 \leq x_l^M \leq 1$, where $x_l^M$ is the fraction of element $l$ mixing on site $M$.

Here, we present a general reformulation of complex non-ideal solution phases from the thermodynamic database of Holland et al. (2018), which comprises equations of state for multiple mineral solid solutions appearing in magmatic systems, as well as multicomponent silicate melt and aqueous fluid phases. Using a nullspace approach, inequality constraints governing the site fractions are transformed into equality constraints, and the resulting problem is turned into an unconstrained optimization problem, subsequently optimized using efficient gradient-based methods. To test our formulation, we apply it to several equations of state for solution phases known for their complexity and compare the results of our approach against classical optimization algorithms supporting inequality constraints.

We find that the BFGS algorithm yields by far the best performance and stability with respect to the other investigated methods, improving the minimization time of individual solution phase by a factor $\geq 10$. We estimate that our new approach can improve the computational time of stable phase equilibrium by a factor $\geq 5$, thus potentially allowing to model realistic reactive fluid/magmatic systems by directly integrating phase equilibrium calculations in multiphase thermomechanical codes.





# 1 Introduction

While the last decade has seen significant progress in thermomechanical modelling of complex multiphase systems (e.g., Keller et al., 2013; Taylor-West and Katz, 2015; Keller and Katz, 2016; Keller et al., 2017; Turner et al., 2017; Keller and Suckale, 2019; Rummel et al., 2020; Katz et al., 2022), the coupling with petrological modelling, when addressed at all, remains largely simplified (Riel et al., 2019). There are two key obstacles. First, most phase equilibrium modeling tools (e.g., Perple_X, Theriak_Domino, geoPS, MELTS (Connolly, 2005; de Capitani and Petrakakis, 2010; Xiang and Connolly, 2021;

Ghiorso and Sack, 1995) have been developed with the primary aim of producing phase diagrams and do not offer useful interfaces to integrate with (parallel) geodynamic codes. Second, phase equilibrium modelling is generally achieved by solving a Gibbs energy minimization problem which is computationally challenging. Several numerical strategies have been developed to solve such optimization problems (Ghiorso and Sack, 1995; Connolly, 2005; de Capitani and Petrakakis, 2010; Piro, 2011; Xiang and Connolly, 2021) and some of the most efficient algorithms rely on repeated solution model minimization in order to

compute for the most stable mineral assemblage (e.g., de Capitani and Petrakakis, 2010; Xiang and Connolly, 2021; Riel et al., 2022). Although computational performance have been significantly increased over the past few years (Xiang and Connolly, 2021; Riel et al., 2022), single point equilibrium prediction is still costly, with computational times of the order of 10s to 100s of milliseconds (e.g., Riel et al., 2022). This limitation effectively precludes direct coupling of phase equilibrium calculations with thermomechanical models, which requires performing 1000s to 100'000s of such calculations every timestep.

In order to account for chemical separation in geodynamic models, several computationally cheaper workarounds have been used. This includes the use of pre-computed set of pseudosections (e.g., Magni et al., 2014; Bouilhol et al., 2015; Rummel et al., 2020) and parameterizations (e.g., Jackson et al., 2003, 2018; Hu et al., 2022; Keller et al., 2022). In (Rummel et al., 2020), the authors generated a database of pre-computed results from phase-equilibria modelling covering the explored/expected compositional, pressure and temperature range of the system. While this approach is powerful, it suffers several limitations.

First, to generate a relevant petrological database, the geodynamic model has to be run multiple times in order to characterize the effective pressure, temperature and compositional range of the system. Second, the database is by definition discrete which implies that a compositional tolerance has to be applied when computing the stable phase equilibrium, thus leading to mass conservation issues.

Although the heavy computational requirements of stable phase equilibrium modelling remains a major obstacle for direct

coupling, recently developed minimization tools yielded a significant improvement in performance (Xiang and Connolly, 2021; Riel et al., 2022). The recent performance increase mainly results from combining/improving existing minimization methods and making use of gradient-based minimization of individual phases to speed up the computations. Several gradient-based minimization methods are currently employed in the different routines computing phase equilibria. Theriak-Domino (de Capitani and Brown, 1987; de Capitani and Petrakakis, 2010) uses either steepest gradient or Newton-Raphson methods.

Minimization of the solution phase model is achieved using a feasible starting guess and continues until a bound or a site





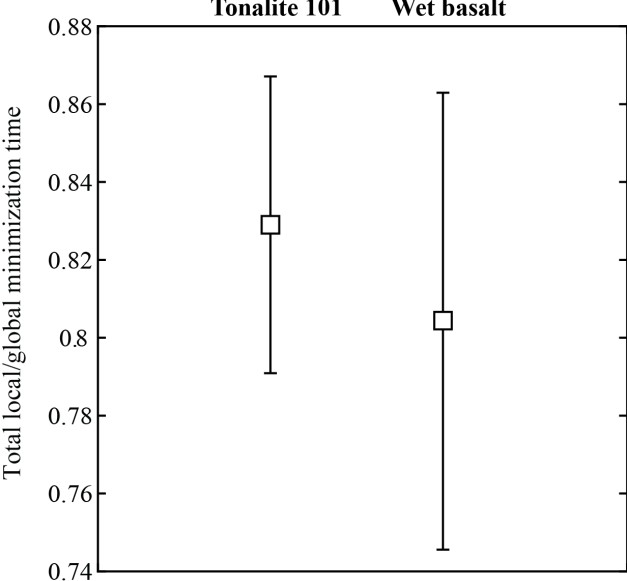

**Figure 1.** Ratio of time spent solving the local problem of the equilibrium composition and order in individual solution models ("total local"), to time spent solving the entire problem of establishing which is the most stable of the possible phase equilibria ("global minimization"), using MAGEMin (Riel et al., 2022) for two representative test cases. Both tonalite and wet basalt bulk-rock compositions are taken from (Holland et al., 2018). In total 619 points were computed from 0 to 12 kbar and from 600 to 900°C and from 800 to 1100°C for the tonalite and the wet basalt case, respectively.

fraction constraint is violated. In our recent phase equilibrium calculation software MAGEMin (Riel et al., 2022), the analytical expressions of the equations of state for solution phases are passed to NLopt software package(Johnson, 2021). Subsequently, the objective function is minimized using the CCSAQ algorithm (Svanberg, 2002) which solves for inequality-constrained nonlinear programming problems. During the inner iterations, a series of convex sub-problems approximating the objective

function and the constraints are generated and solved until the constraints are satisfied (Svanberg, 2002). This procedure is repeated until the solution phase model is minimized. geoPS (Xiang and Connolly, 2021) uses the simulated annealing (SA) method. Compared to gradient-based methods, simulated-annealing is a probabilistic technique for approximating the global optimum of a given function (e.g., Pincus, 1970). Here, constraints can be accounted for as penalties on the objective function. In the method of (Riel et al., 2022), inequality constraints arise from the requirement that the site fractions in the

equations of state for individual solution phases (xeos) should retain physical values, in the range 0–1, as discussed below. The first release of MAGEMin followed the THERMOCALC software (Powell and Holland, 1988) in treating the constraint of physicality of site fractions as a set of inequality constraints on the solution of an equilibrium, in which the values of the site fractions in the solution phases present must all be $\geq 0$. This also ensures that all site fractions are $\leq 1$, since the parameterisation of site fractions forces the total of all site fractions associated with a given site to be unity. The use of

inequality constraints gradient-based methods, results in relatively slow performances and occasional solver failure due to





slight violation of inequality constraints. Using the first publicly released version of MAGEMin (Riel et al., 2022), we find that the global minimization time is largely dominated by how fast gradient-based minimization of individual solution phases can be performed, with 75 to 90% of the computation time dedicated to local minimization to find the equilibrium compositions and state of order of solution models (see figure 1). Therefore, it becomes critically important to improve the minimization time of individual solution phase models to further speed-up the overall phase-equilibrium computational time.

Here, we present a revised implementation of the compositional variables (xeos) of (Holland et al., 2018) within MAGEMin that avoids to express the site fraction as inequality constraints. Elimination of these constraints allows using faster unconstrained optimization methods, thus considerably improving performance and stability of the code. We compare the accuracy and performance of two well-known unconstrained gradient-based optimization methods: the conjugated gradient (CG) and the Broyden-Fletcher-Goldfarb-Shanno (BFGS) method.

## 2 Methodology

### 2.1 Solution phase formulation

At fixed pressure $P$ and temperature $T$, the total Gibbs energy of solution phase $\lambda$ is given by

$$G_\lambda = \sum_{i=1}^{N_\lambda} \mu_{i(\lambda)} p_{i(\lambda)}. \tag{1}$$

where $N_\lambda$ the number of end-members of solution phase $\lambda$, $p_{i(\lambda)}$ is the fraction of end-member $i$ dissolved in solution phase $\lambda$ and $\mu_{i(\lambda)}$ is the molar chemical potential of end-member $i$ in solution phase $\lambda$. An end-member is defined as an independent instance of a solution phase, at a single specified composition, for which the Gibbs energy is fully defined as a function of pressure and temperature only. In a given chemical system, the linear combination of the end-members span the complete crystallographic site-occupancy space of the solution phase.

The chemical potential of a phase is a function of the dissolved end-members within a solution phase (see Ganguly, 2001, for a review)

$$\mu_{i(\lambda)} = g^0_{i(\lambda)} + RT \log(a^{id}_{i(\lambda)}) + g^{\mathrm{ex}}_{i(\lambda)}, \tag{2}$$

where $R\ [\mathrm{Jmol^{-1}K^{-1}}]$ is the ideal gas constant, $T$ [K] is the absolute temperature, $a^{id}_{i(\lambda)}$ is the ideal activity coefficient, $g^0_{i(\lambda)}$ the Gibbs energy of reference of the pure end-member Helgeson (1978); Holland and Powell (1998) and $g^{\mathrm{ex}}_{i(\lambda)}$ is the excess energy term Powell and Holland (1993); Holland and Powell (2003). The ideal activity coefficient $a^{id}_{i(\lambda)}$ is generally defined as $a^{id}_{i(\lambda)} = p_{i(\lambda)}$ for molecular mixing, or else for mixing on crystallographic sites as

$$a^{id}_{i(\lambda)} = c_i \prod_s (X^s_{e_{s,i}})^{\nu_s} \tag{3}$$

where $X^s_{e_{s,i}}$ is the site fraction of the element $e_{s,i}$ that appears on site $s$ in end-member $i$ of phase $\lambda$, $\nu_s$ is the number of atoms contained in mixing site $s$ of $\lambda$, and $c_i$ is a normalisation constant that ensures that $a^{id}_{i(\lambda)}$ is unity for the pure end-member $i$.





In the asymmetric formalism, $g_i^{ex}$ is given by:

$$g_{i(\lambda)}^{ex} = - \sum_{m=1}^{N_{ol}-1} \sum_{n>m}^{N_{ol}} (\phi'_m - \phi_m)(\phi'_n - \phi_n) W_{m,n} \left( \frac{2v_i}{v_m + v_n} \right), \qquad (4)$$

where $\phi_i$ is the proportion of end-member $i$ weighted by the asymmetry parameters, as $\phi_i = (p_i v_i)/(\sum_{m=1}^{N_{ol}} p_m v_m)$, with $v_i$ the asymmetry parameter for end-member $i$. $\phi'_m$ is the value of $\phi_m$ in end-member $i$, such that $\phi'_m = 1$ where $m = i$ and $\phi'_m = 0$ where $m \neq i$. $W_{m,n}$ is the interaction energy between end-members $m$ and $n$ in the solution.

In Holland et al. (2018), composition (the overall ratios of elements) and order (the distribution of elements over mixing sites) in an xeos are parameterised in terms of an independent set of variables (see example below). Given this formulation, the set of equations 1 to 4 can be directly transformed into the following optimization problem

$$\min G_\lambda(x_{cv}) = \sum_{i=1}^{N_\lambda} \mu_{i(\lambda)} p_{i(\lambda)}, \qquad (5)$$

subject to

$$X_{e_s,i}^s \geq 0, \qquad (6)$$

and

$$\text{lb}_{cv} \leq x_{cv} \leq \text{ub}_{cv}, \qquad (7)$$

where $\mu_{i(\lambda)}$, $p_{i(\lambda)}$, and $X_{e_s,i}^s$ are function of the compositional and order variables $x_{cv}$, and, $\text{lb}_{cv}$ and $\text{ub}_{cv}$ are the lower and upper bounds on the set of compositional and order variables $x_{cv}$. The first derivative of $f(x_{cv})$ is given by

$$\frac{\partial f}{\partial x_{cv}} = \mu_{i(\lambda)} \frac{\partial p_{i(\lambda)}}{\partial x_{cv}}, \qquad (8)$$

and the first derivative of the equality constraints by

$$\frac{\partial X_{e_s,i}^s}{\partial x_{cv}}. \qquad (9)$$

## 2.2 A revised formulation

The solid solutions presented in Holland et al. (2018) are formulated on the basis of exchanging chemical species on a finite number of unique crystallographic sites (Bragg–Williams-type formulation, see Myhill and Connolly (2021) for more details). A key challenge with this formulation is that minimization has to be performed while keeping site fractions $\geq 0$. Our previous implementation (Riel et al., 2022) imposed these inequalities directly, which is numerically significantly more costly than when using unconstrained minimisation algorithms. In order to simplify the optimization problem and improve minimization time, we here propose an alternative formulation that converts inequality constraints into linear equality constraints, as discussed below using olivine as a representative example.



The olivine solid solution model (Holland et al., 2018) contains 2 mixing sites M1 and M2 and represents a phase that can be expressed by the general formula:

$$[\mathrm{Mg^{2+}, Fe^{2+}}]^{\mathrm{M1}}[\mathrm{Mg^{2+}, Fe^{2+}, Ca^{2+}}]^{\mathrm{M2}}\mathrm{SiO_4} \tag{10}$$

Here $\mathrm{Mg^{2+}}$ and $\mathrm{Fe^{2+}}$ can be exchanged on crystallographic site M1 and $\mathrm{Mg^{2+}}$, $\mathrm{Fe^{2+}}$ and $\mathrm{Ca^{2+}}$ can be exchanged on site crystallographic M2. In Holland et al. (2018), the site fractions are expressed as (dropping the charges in the notation for the ions):

$$X_{\mathrm{Mg}}^{\mathrm{M1}} = 1 - x + Q \tag{11}$$
$$X_{\mathrm{Fe}}^{\mathrm{M1}} = x - Q \tag{12}$$
$$X_{\mathrm{Mg}}^{\mathrm{M2}} = (1-x)(1-c) - Q \tag{13}$$
$$X_{\mathrm{Fe}}^{\mathrm{M2}} = x(1-c) + Q \tag{14}$$
$$X_{\mathrm{Ca}}^{\mathrm{M2}} = c \tag{15}$$

where composition has been parameterised using the variables $x = (X_{\mathrm{Fe}}^{\mathrm{M1}} + X_{\mathrm{Fe}}^{\mathrm{M2}})/(X_{\mathrm{Fe}}^{\mathrm{M1}} + X_{\mathrm{Fe}}^{\mathrm{M2}} + X_{\mathrm{Mg}}^{\mathrm{M1}} + X_{\mathrm{Mg}}^{\mathrm{M2}})$ and $c = X_{\mathrm{Fe}}^{\mathrm{M1}}$, and order has been parameterised using the variable $Q = x - (X_{\mathrm{Fe}}^{\mathrm{M1}})/(X_{\mathrm{Fe}}^{\mathrm{M1}} + X_{\mathrm{Mg}}^{\mathrm{M1}})$. This parameterisation ensures that the site fractions on each of the individual sites are inherently normalised to 1. Two other types of constraint might be built into the parameterisation in a more complex example: (i) charge balance: if variably-charged ions were mixing, charge balance would be maintained during compositional change, and (ii) equidistribution: the xeos might be simplified by equating two site fractions, typically involving minor elements. The resulting set of composition and order variables is an independent set, that fully and uniquely describes the site occupancies at a given composition and state of order, subject to physical constraints arising from the lattice structure of the mineral. The relationship between the number of composition and order variables and the number of site fractions is then given by:

$$n_{sf} = n_{x_{cv}} + n_{eq_{norm}} + n_{eq_{cb}} + n_{eq_{edist}}, \tag{16}$$

with $n_{sf}$ the number of site fractions, $n_{x_{cv}}$ the number of compositional and order variables, $n_{eq_{norm}}$ the number of constraints arising from the normalisation of site fractions on a given site to 1, $n_{eq_{cb}}$ the number of charge balance equations, equal to 0 or 1, and $n_{eq_{edist}}$ the number of equidistribution constraints imposed. Collectively, the normalised charge balance and equidistribution constraints form a set of linear equalities among the site fractions.





In the revised implementation, we retrieve the set of equality constraints from the site fractions for olivine as follows. We first take the partial derivatives of site fractions as functions of the compositional and order variables as:

$$\frac{\partial X^s_{e_{s,i}}}{\partial x_{cv}} = \begin{bmatrix} -1 & 0 & 1 \\ 1 & 0 & -1 \\ c-1 & x-1 & -1 \\ 1-c & -x & 1 \\ 0 & 1 & 0 \end{bmatrix}, \tag{17}$$

where $x = X^s_{e_{s,i}}$ is the site fraction of the element $e_{s,i}$ that appears on site $s$ and $x_{cv}$ the set of compositional and order variables

$cv$. Next, we compute the set of linear constraints using symbolic expressions as

$$A = \text{Null}\left(\left(\frac{\partial X_e}{\partial x_{cv}}\right)^T\right) = \begin{bmatrix} 1 & 1 & 0 & 0 & 0 \\ 0 & 0 & 1 & 1 & 1 \end{bmatrix}, \tag{18}$$

where $A$ is the set of equality constraints imposed on the site fractions by the parameterisation of the xeos, and Null stands for the null space. Here, each of the $p$ rows of $A$ represents an equality constraint, while the value in column $q$ represents the coefficient applied to the $q^{\text{th}}$ site fraction in the constraint. As expected, this returns set of linear equalities on the olivine site

fractions, comprising two site normalisation expressions:

$$1 \times \text{xMgM1} + 1 \times \text{xFeM1} = 1.0, \tag{19}$$

and

$$1 \times \text{xMgM2} + 1 \times \text{xFeM2} + 1 \times \text{xCaM2} = 1.0. \tag{20}$$

Note that, for more complex xeos and depending on the arbitrary order in which the site fractions are listed, the nullspace

operation may not yield expressions that are straightforward statements of the site normalisation, charge balance and equidistribution constraints. However, it will always yield an independent set of linear equalities of length $n_{eq_{norm}} + n_{eq_{cb}} + n_{eq_{edist}}$, which are exactly equivalent to, and can be linearly recombined to give such straightforward statements.

In the implementation of Riel et al. (2022), MAGEMin solved for the equilibrium composition and state of order of a phase in terms of the variables $x_{cv}$, subject to the constraint that the values of site fractions should be $\geq 0$. To eliminate the site fraction

inequalities from the implementation, we now wish to solve directly for the site fractions, while subjecting them to the equality constraints obtained via the nullspace operation. The resulting problem can be expressed as

$$\min f(x), \tag{21}$$

subject to

$$Ax = b, \tag{22}$$





where $x$ represent the $n$ site-fractions and $Ax = b$ is the set of $p$ equality constraints (equation 18).

The linear equality constraints can now be eliminated from the problem, reducing the number of variables solved for back to $n_{x_{cv}}$, the number of variables needed to uniquely describe the composition and state of order. This is done by finding a matrix $F \in R^{n-p}$ and a vector $\hat{x} \in R^n$ that parameterize the feasible set (Boyd et al., 2004) such as

$$\left\{ x \mid Ax = b \right\} = \left\{ N_z + \hat{x} \mid z \in R^{n-p} \right\}, \tag{23}$$

where $N_z$ is a null space matrix of $A$ and $\hat{x}$ is any particular solution of $Ax = b$. A particular solution and the null space can both be found by doing a full $QR$ decomposition on $A$ such that:

$$A = Q \begin{bmatrix} R_1 \\ 0 \end{bmatrix} = \begin{bmatrix} Q_1 \ Q_2 \end{bmatrix} \begin{bmatrix} R_1 \\ 0 \end{bmatrix}, \tag{24}$$

where the vector $q^{n+1}, \ldots, q^m$ are an orthonormal basis for the null space of $A^T$ and solutions to the set of equality constraints can be obtained using

$$\hat{x} = Q_1 {R_1}^T b + Q_2 z, \tag{25}$$

where the $Q_1 {R_1}^T b$ is a particular solution and $Q_2 z$ gives a vector in the null space.

In other words, the use of the null space of $A$ ($N_z$) parameterizes the space such that for any step $\Delta z$, $\hat{x} + F z \Delta z$ remains in the feasible domain.

Using the elimination method, equation 1 becomes

$$\min f(x) = G_\lambda(X_{e_{s,i}}^s) = \sum_{i=1}^{N_\lambda} \mu_{i(\lambda)} p_{i(\lambda)}, \tag{26}$$

and the parameterized first derivative becomes

$$\frac{\partial f}{\partial x}(z) = \frac{\partial G_\lambda}{\partial X_{e_{s,i}}^s}(z) = N_z \left( \left( \frac{\partial G_\lambda}{\partial x} \right)^T N_z \right)^T, \tag{27}$$

where $x = X_{e_{s,i}}^s$ is the site fraction of the element $e_{s,i}$ that appears on site $s$. Equation 26 is then minimized using the gradient information given by 27 and the methods presented below.

## 2.3 Gradient-type iterative methods

Considering the unconstrained optimization problem

$$\min f(x), \tag{28}$$

where $f(x)$ is twice continuously differentiable. The general gradient-type iterative method to solve this problem is of the form

$$x_{k+1} = x_k + \alpha_k d_k, \tag{29}$$





for iteration $k \geq 0$, where $d_k$ is the search direction and $\alpha_k$ is the step-length. In this study, we compute the step-length using a Wolfe line search (Wolfe, 1969) such that the inequalities

$$f(x_k + \gamma_k \alpha_k d_k) \leq f(x_k) + \rho \gamma_k \alpha_k g_k^T d_k, \tag{30}$$

and

$$g_{k+1}^T d_k \geq \sigma g_k^T d_k, \tag{31}$$

are satisfied, where $0 < \rho < \sigma < 1$ and $\gamma_x$ is the maximum feasible step-length computed as

$$\gamma_x = \begin{cases} 1 / \frac{\min(x_{k+1} - \epsilon)}{\text{abs}(d_k)}, & \text{if any } x_k + d_k \leq 0 \\ 1, & \text{otherwise,} \end{cases} \tag{32}$$

where $\epsilon$ is a small number, typically $\leq 10^{-8}$. The maximum feasible step-length $\gamma_x$ ensures that the values of site-fractions remain $\geq \epsilon$. Iterations are then processed until a stopping criterion is satisfied. Because solution phases are not necessarily convex during global Gibbs energy minimization, we set the stopping criteria using the relative change of the objective function. The stopping criteria is met when

$$\text{abs}((f_k - f_{k-1})/f_{k-1}) < \text{tol}, \tag{33}$$

where $\text{tol}$ is a small number typically $\leq 10^{-8}$.

If the descent direction $d_k$ is simply chosen to be $d_k = -g_k$ we obtain the steepest descent algorithm. However, this approach is known to be prone to oscillation (e.g., Nocedal et al., 2002) and slow convergence, and will therefore not be explored. Instead, we test two unconstrained optimization methods that use the gradient information of the previous iteration(s), namely the conjugate gradient and the Broyden-Fletcher-Goldfarb-Shanno (BFGS) method.

### 2.3.1 Conjugate Gradient method

For the conjugate gradient method, the descent direction is defined by

$$d_k = -g_k, \tag{34}$$

if the iteration increment k = 0, or

$$d_k = -g_k + \beta_k d_{k-1}, \tag{35}$$

for increment k $\geq 1$. Here, $g_k$ is the gradient $g(x)$ of function $f(x)$ at point $x_k$, and $\beta_k$ is the conjugate gradient update parameter. Variants of the conjugate gradient method are defined by using different update parameters $\beta_k$ (see for example Hestenes and Stiefel, 1952; Rivaie et al., 2012, 2015). Here, we employ the three-term conjugate gradient method presented by Liu et al. (2018) with the update parameter $\beta_k$ defined in Rivaie et al. (2015)

$$\beta_k = \frac{g_k^T (g_k - g_{k-1} - d_{k-1})}{\|d_{k-1}\|^2}, \tag{36}$$





and further extend the descent direction term as

$$d_k = \text{-}g_k + \beta_k d_{k-1} + \theta_k y_{k-1}, \tag{37}$$

where $y_{k-1} = g_k - g_{k-1}$ and

$$\theta_k = -\frac{g_k^T d_{k-1}}{\|d_{k-1}\|^2}. \tag{38}$$

A useful property of the three-term conjugate gradient method is that the search direction always satisfies the sufficient descent condition without any line search (Liu et al., 2018).

The descent direction is parameterized to satisfy the equality constraints (equation 18) such as

$$d_k^p = N_z(d_k^T N_z)^T. \tag{39}$$

### 2.3.2   BFGS method

The Broyden-Fletcher-Goldfarb-Shanno (BFGS) method is a well-known quasi-Newton method for solving unconstrained optimization problems (see for instance Fletcher, 1987; Dennis Jr and Schnabel, 1996). The quasi-Newton descent direction is given by

$$d_k = -B^{-1}g_k, \tag{40}$$

where $B^{-1}$ is the inverse of the Hessian matrix. Here, we approximate $B^{-1}$ using the Sherman-Morrison formula (Sherman and Morrison, 1950) such as

$$B_{k+1}^{-1} = B_k^{-1} + \frac{(s_k^T y_k + y_k^T B_k^{-1} y_k)(s_k s_k^T)}{(s_k^T y_k)^2} - \frac{B_k^{-1} y_k s_k^T + s_k y_k^T B_k^{-1}}{s_k^T y_k} \tag{41}$$

where $s_k = x_{k+1} - x_k$, $y_k = g_{k+1} - g_k$ and $B_{k=0}^{-1}$ is initialized with the identity matrix.

Because of the relatively low dimensionality of the solution phase model ($< 20$) we do not consider the limited-memory BFGS method (L-BFGS) and instead update $B_k^{-1}$ as shown in equation 41 during every iteration. Once the problem has converged (i.e. equation 33 is satisfied), we reset the Hessian matrix inverse $B_{k-1}$ to the identity matrix and perform additional iteration(s). This ensures that the problem converges to its local minimum in the event the quality of the approximate Hessian matrix inverse $B_{k-1}$ is degraded.

As for the conjugate gradient method, the descent direction is parameterized to satisfy the equality constraints (equation 18) such as

$$d_k^p = N_z(d_k^T N_z)^T. \tag{42}$$

## 3   Application

In order to test the unconstrained solution phase formulation, we selected, from the Holland et al. (2018) set of xeos, three
solution phases with complex features including high-dimensional composition–order spaces and geologically significant solvi:



clinoamphibole (NCKFMASHTO), clinopyroxene (KNCFMASTOCr) and spinel (FMATOCr). We use as starting points the set of feasible points of each discretized solution phase. Discretization of the solution phases is achieved using a compositional variable step of 0.25 which yielded 5498, 4124 and 1521 feasible starting points (or pseudocompounds) for clino-amphibole, clinopyroxene and spinel, respectively. Because gradient-based minimization of solution phase models is achieved with respect

to a given Gibbs hyperplane (de Capitani and Brown, 1987; de Capitani and Petrakakis, 2010; Xiang and Connolly, 2021; Riel et al., 2022), we first compute the phase equilibrium at a given pressure, temperature and bulk-rock composition to retrieve the global minimum Gibbs hyperplane using MAGEMin (Table 1). All computations were performed on a Linux (x86_64-linux-gnu) operating system, utilizing a 6-core 11th Gen Intel(R) Core(TM) i5-11400H CPU running at 2.70 GHz.

**Table 1.** Solution phase models parameters.

| Tested phase | **clino-amphibole** | **clinopyroxene** | **spinel** | **spinel solvus** |
|---|---|---|---|---|
| Pressure [kbar] | 5.0 | 12.0 | 12.0 | 3.26 |
| Temperature [C] | 650.0 | 1100.0 | 1100.0 | 906.25 |
| Number of points | 4950 | 4121 | 1521 | 1521 |
| Tested methods | CCSAQ, SLSQP, CG, BFGS | CCSAQ, SLSQP, CG, BFGS | CCSAQ, SLSQP, CG, BFGS | BFGS |
| Number of dimensions | 17 | 13 | 10 | 10 |
| Oxides | Reference Gibbs hyperplane [J] | | | |
| $SiO_2$ | -960.9655 | -1011.909631 | -1011.909631 | -1001.730935 |
| $Al_2O_3$ | -1768.2476 | -1829.092564 | -1829.092564 | -1818.611331 |
| CaO | -788.4474 | -819.264126 | -819.264126 | -812.972365 |
| MgO | -678.9683 | -695.467358 | -695.467358 | -689.113013 |
| FeO | -355.2975 | -412.948568 | -412.948568 | -396.911228 |
| $K_2O$ | -914.9708 | -971.890270 | -971.890270 | -966.511310 |
| $Na_2O$ | -839.9561 | -876.544354 | -876.544354 | -882.719670 |
| $TiO_2$ | -1008.3630 | -1073.640927 | -1073.640927 | -1045.994137 |
| O | -263.7269 | -276.590707 | -276.590707 | -249.181839 |
| $Cr_2O_3$ | -1262.6087 | -1380.299631 | -1380.299631 | -1332.815844 |
| $H_2O$ | -368.4674 | - | - | - |

The performance and reliability of the unconstrained formulations are tested against the inequality constrained formulations

using the SLSQP (Kraft, 1988, 1994) and CCSAQ methods (Svanberg, 2002). The minimizations using the SLSQP and CCSAQ methods were computed using the C implementation of NLopt (Jackson et al., 2018) through MAGEMin as described in Riel et al. (2022) and in the scripts provided in supplementary materials. Because the algorithms explored in this study (Julia implementation of CG and BFGS methods) exhibited similar accuracy with residual $\leq 10^{-13}$, the differences in algorithm





accuracy are not discussed. Here, a minimization is considered successful when the norm of the distance to the solution is
$\leq 10^{-4}$.

## 4 Discussion

### 4.1 Algorithms performance and reliability

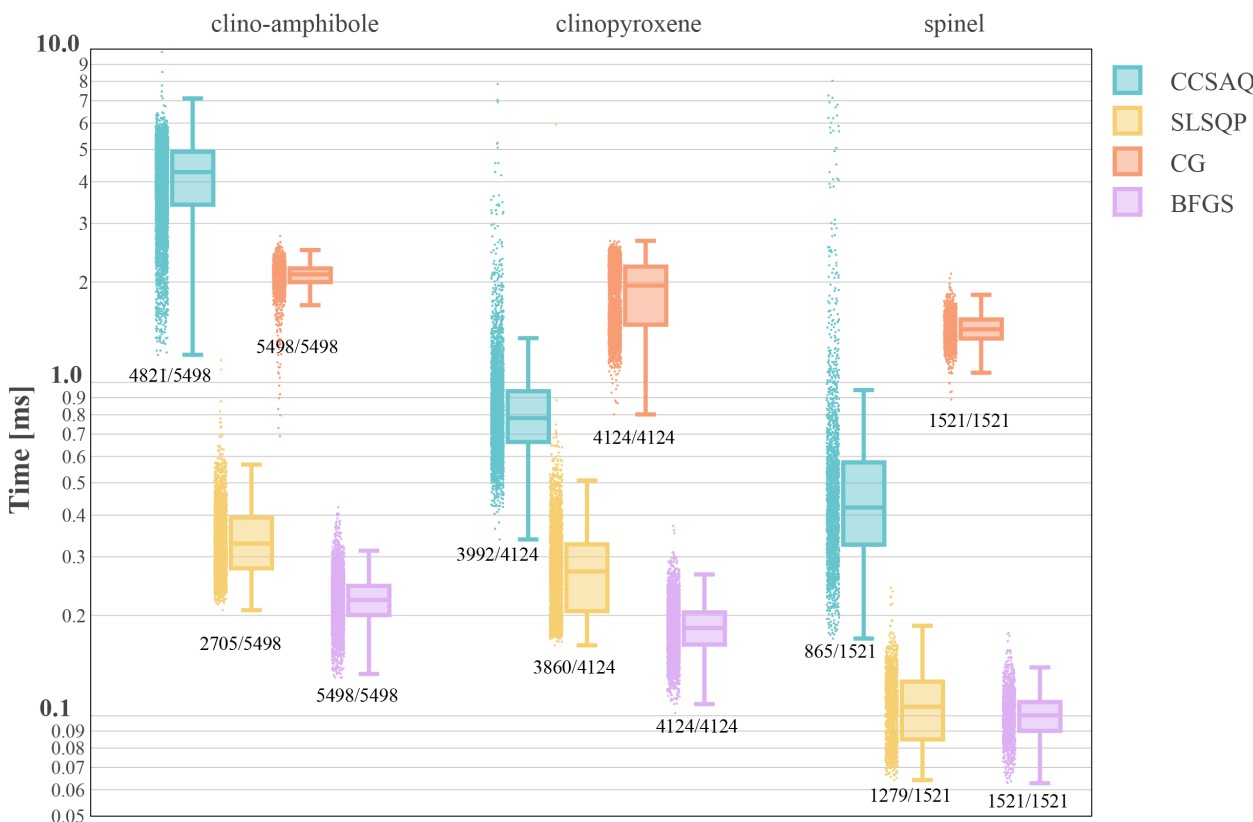

**Figure 2.** Minimization time box plot for tested solution phases and optimization methods. SLSQP, Sequential Levast-Squares Quadratic Programming (supporting both inequality and equality constraints); CCSAQ, Conservative Convex Separable Approximation with Quadratic penalty; CG, conjugated gradiend; BFGS, Broyden-Fletcher-Goldfarb-Shanno. 5498, 4124 and 1521 starting points for clino-amphibole, clinopyroxene and spinel, respectively. Starting points were generated by evenly sampling the entire feasible space following the method present in Riel et al. (2022). The numbers below the boxes show the number of successful minimization over the total number of tested points.

The box plots depicted in Fig. 2 illustrate that the performance of the unconstrained CG method is comparable to that of the inequality-constrained CCSAQ method (implemented via NLopt). While the CG method outperforms CCSAQ for amphibole, 275 with minimization times of ∼2100 $\mu$s versus ∼4200 $\mu$s, respectively, the efficiency of CCSAQ is larger for problems with





lower dimensionality, such as clinopyroxene and spinel. The SLSQP method demonstrates superior efficiency, with average minimization times of ~340 $\mu$s for amphibole, ~270 $\mu$s for clinopyroxene, and ~120 $\mu$s for spinel. However, the BFGS algorithm outperforms SLSQP, achieving average minimization times of ~220 $\mu$s for amphibole, 180 $\mu$s for clinopyroxene, and ~100 $\mu$s for spinel; a performance increase of 20 to 50%. Additionally, the BFGS method's convergence requires between

25 to 90 iterations across different solution phase models, as indicated in Fig. 3. Notably, the minimum time per iteration is influenced by the dimensionality of the solution phase model, ranging from ~4.0 $\mu$s per iteration for clino-amphibole to ~2.1 $\mu$s for spinel (Fig. 3 and table 1).

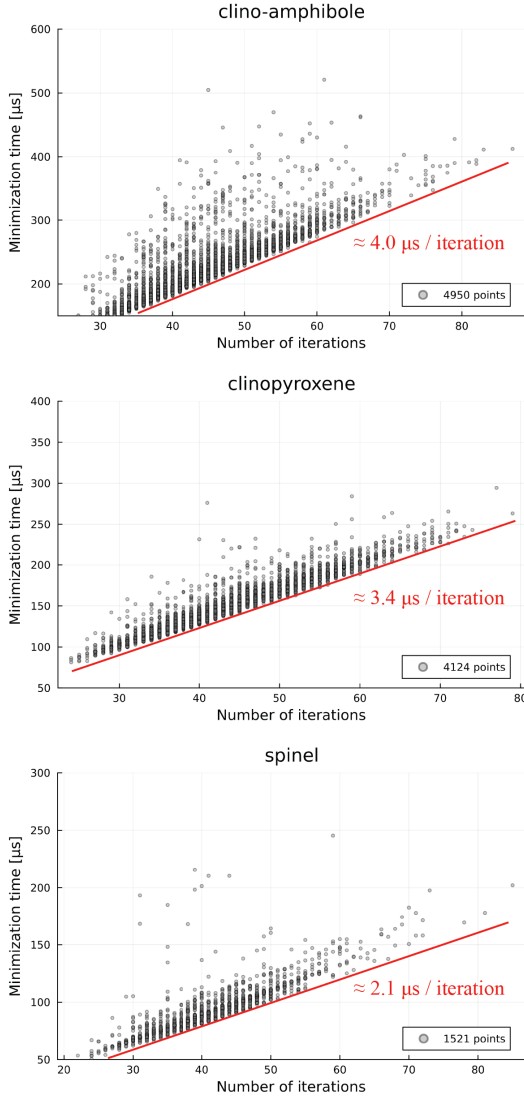

**Figure 3.** Number of iterations versus minimization time for the BFGS method. The red lines show the minimum minimization time per iteration.

Although the average minimization time is a good indicator of the raw performance of the algorithms, reliability of the solvers is of key importance when computing phase equilibria. In this light, we find that the unconstrained algorithms (CG and BFGS) are far superior to the inequality constraints ones (CCSAQ and SLSQP). For instance, the unconstrained methods (BFGS and CG) successfully minimize 100% of the tested starting points (Fig. 2) while the inequality constraints methods show a significant amount of unsuccessful minimization reaching up to 50% in some cases e.g., clino-amphibole minimization using SLSQP or spinel using CCSAQ (Fig. 2). The unsuccessful minimizations are related to violated inequality constraints and the inability for the algorithms (SLSQP and CCSAQ) to go back to the feasible domain.

Finally, we tested the BFGS algorithm for an equilibrium between two phases separated by a solvus i.e., an objective function containing more than one local minimum. The parameters of the test are given in table 1 (spinel solvus) and the results are shown in figure 4. The unconstrained formulation and the BFGS method perfectly captures the solvus with consistent minimization time similar to those shown in figure 2

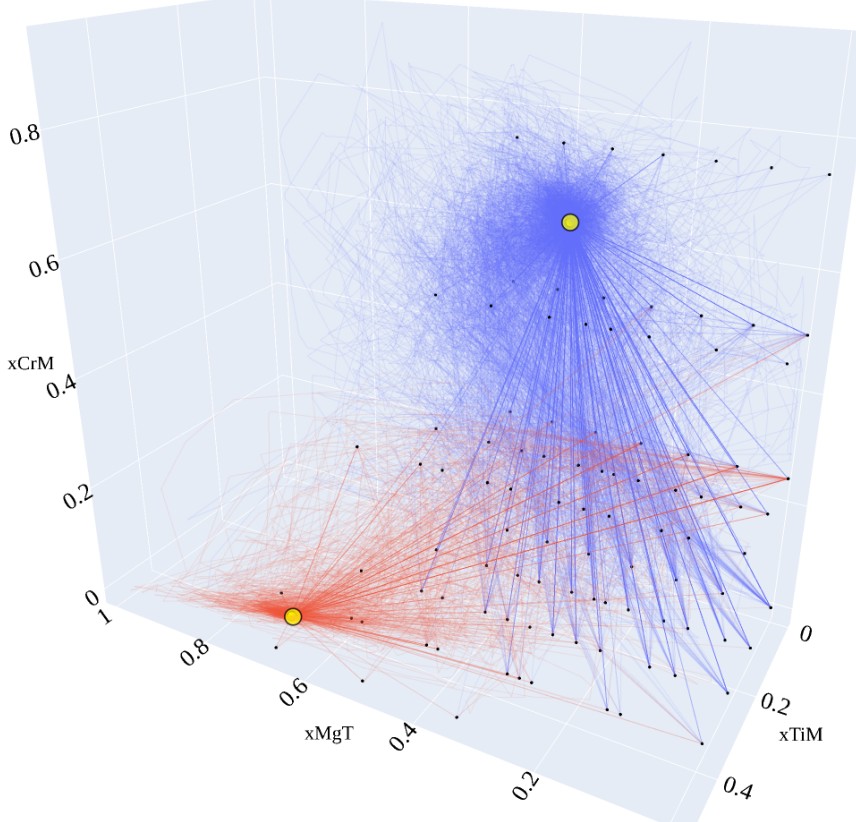

**Figure 4.** Minimization paths for spinel solvus which has two local minima. Black dots, starting point in the xCrM, xMgT and xTiM site fraction sub-system; yellow circles with black outline, local minimum of the solvus; red and blue lines, minimization paths from starting point to local minimum. Note that the diagram only displays a 3D sub-system of the full 10D system.





## 4.2 Minimization of perturbed systems

Minimization from discretized starting points allows quantification and comparison of the raw performance and stability of the algorithms (CG, BFGS and CCSAQ, see Fig. 2). However, a phase equilibrium calculation employing gradient-based methods generally involves finding a new local minimum under slightly to moderately perturbed conditions between global iterations (de Capitani and Petrakakis, 2010; Riel et al., 2022) i.e., that the distance between the starting point and the ending point during local minimization of individual phases is small ($\|\Delta\Gamma\|_2^2 < 10.0$). In order to test the performance of the SLSQP and BFGS

methods under small perturbations, we use as starting points the minima obtained by tests 1 to 3 and apply a perturbation to the Gibbs hyperplane (table 1). The perturbation is set by applying a random rotation to the objective function which shifts the local minimum from its current position. We explore the effect of such perturbation by computing 10'000 random rotations per solution phase yielding a range of chemical potential $\|\Delta\Gamma\|_2^2$ varying from 0.0 to ca. 60.0. The results of the minimizations of the rotated systems are presented in figure 5A to C.

For perturbed conditions, we find that the minimization time of the SLSQP algorithm does not scale with the norm of the perturbation ($\|\Delta\Gamma\|_2^2$) (Fig. 5A,B,C). This relationship is independent of the tested solution phase model (Fig. 5A,B,C). Instead, for the BFGS algorithm, the minimization time scales with the norm of the perturbation (Fig. 5D,E,F). For a $\|\Delta\Gamma\|_2^2 < 10.0$ the minimization time is divided by a factor of ca. 2.0 to 3.0 with respect to the mean raw minimization time (Fig. 2) resulting in an average time of 70-80 $\mu$s for clino-amphibole, 50-60 $\mu$s for clinopyroxene and 30-40 $\mu$s for spinel.



**Figure 5.** Minimization time for perturbed systems. A-C SLSQP algorithm applied to clino-amphibole, clinopyroxene and spinel, respectively. D-F, BFGS algorithm applied to clino-amphibole, clinopyroxene and spinel, respectively. $\|\Delta\Gamma\|_2^2$ is a measure of the distance between the starting guess and the solution. Note that the inequality constrained SLSQP method does not show any correlation between the distance to solution and the minimization. The BFGS method, instead, exhibits a clear trend of decreasing minimization for smaller distances to solution.





We propose a revised xeos implementation using a nullspace approach, that allows using unconstrained, rather than constrained, gradient-based optimisation methods. We tested the performance and computational reliability of different algorithms and find that the BFGS method yields the best performance, decreasing the minimization time of individual solution phases by a factor $\geq 10$ compared to CG and CCSAQ methods and by a factor of 1.5 to 2.0 with respect to the SLSQP method (Fig. 2). Under slight perturbations, the minimization time is further decreased by a factor of 2.0 reaching down to $\leq 100$ $\mu$s for
clino-amphibole, $\leq 80$ $\mu$s for clinopyroxene and $\leq 50$ $\mu$s for spinel.

Regarding computational reliability, unconstrained optimization methods(such as CG and BFGS), are clearly preferable over methods with inequality constraints (CCSAQ and SLSQP), which exhibit a considerable proportion of unsuccessful minimizations.

Using an unconstrained nullspace formulation and the BFGS method, can therefore significantly improve the performance of
stable phase calculations. Given that in MAGEMin, $\geq 75\%$ of the computational load is dedicated to local minimization of solution phase models, we estimate a general speed-up of stable phase equilibrium prediction by a factor $\geq 3$. Such improvement can potentially opens up the possibility of performing 2D reactive models fluid/magma transport.

*Code availability.* All the Julia scripts and data necessary to reproduce the results and figures of this contribution are provided on Zenodo (10.5281/zenodo.13982544) and Github on https://github.com/ComputationalThermodynamics/SandBox

*Author contributions.* NR developed and implemented the new formulation, made the analysis and wrote the paper. BK, EG, AdM, and VM provided technical support and feedback during development. HD did the additional testing and improvement of the code. All authors actively contributed to the project through regular meetings and provided critical feedback. All authors revised the final version of the paper.

*Competing interests.* At least one of the (co-)authors is a member of the editorial board of Geoscientific Model Development.

*Acknowledgements.* This study was funded by the European Research Council through the MAGMA project, ERC Consolidator Grant #771143. N.R., B.K. and E.M. would also like to acknowledge the German Research Foundation (DFG) - Project number 521637679 for financial support.



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
