# Peer review of "A bound-constrained formulation for complex solution phase minimization"

_Geoscientific Model Development, 2024_

## Community Comment (CC1)

Review of Riel et al., GMD-2024-197

This is plainly a useful and important contribution. The essential motivation and results are clear. Nonetheless, before publication some minor revisions are needed to make the presentation as clear and unambigious as it can be.

Lines 40-48: People are starting to explore a new cheap work-around, namely neural net emulators of thermodynamic models. See, for example,

https://agu.confex.com/agu/agu24/meetingapp.cgi/Paper/1708197 ... just an abstract at this point, but maybe worth mentioning.

Line 70: Fix grammar: "...inequality constraint in gradient-based methods results in..."

Line 77: "...avoids the need to express ..."

Line 79: Just "conjugate", not "conjugated"

Line 88: "combinations", plural

Equation 2:  $a_{i(\lambda)}^{id}$  as defined here is NOT an "ideal activity coefficient". It is an "ideal activity". I think everybody agrees that the activity coefficient defines the deviation from ideality and is given by  $\gamma_{i(\lambda)}^{id}$  such that  $g_{i(\lambda)}^{ex} = RT\log(\gamma_{i(\lambda)}^{id})$ . A "coefficient" is a multiplicative quantity that modifies some variable term in an equation. So, for example in the polynomial  $f(x) = a + bx + cx^2$ , the coefficients are a, b, and c. The term "bx" is not a coefficient. Also, better specify, because the usage is not universal, that here "log" means natural logarithm.

Line 113: "functions"

Line 138: c is defined incorrectly. Obviously, it is equal to  $X_{Ca}^{M2}$ , not  $X_{Fe}^{M1}$  !

Equation 17: The meaning of the Jacobian matrix is vague unless you explicitly spell out the order of the terms in the vectors, i.e.  $\mathbf{X}_{e_s}^s = \{X_{Mg}^{M1}, X_{Fe}^{M1}, X_{Mg}^{M2}, X_{Fe}^{M2}, X_{Ca}^{M2}\}$  and  $\mathbf{x}_{cv} = \{x, c, Q\}$ .

Equations 17-18: make up your mind about notation. Is the numerator in the Jacobian matrix and in the null-space expression  $X_{e_{s,i}}^s$ , x, or  $X_e$ ? Recommend you use a bold character since it is really a vector.

Equation 21: Again, make up your mind: x or  $x_{cv}$ ?

Equation 22: What is *b*? Here it appears to be  $\{1, 1\}$ , but this is never stated. Equation 18 is not the set of equality constraints, it is just the definition of the matrix *A* (which, again, ought to have a distinctive symbol to indicate it is a matrix).

Line 183: It is not obvious (to those that don't do QR decompositions for a living) that Q is a matrix whose columns are  $q^1 \dots q^m$ , or that these are ordered into  $Q_1$  and  $Q_2$  where  $Q_1$  has n columns corresponding to the image and  $Q_2$  has m-n columns corresponding to the null space. Please say that.

Do you really need to introduce the matrix F? Isn't it just  $Q_2$ ? Or for that matter,  $N_z$  as the nullspace matrix of A when A is already the nullspace matrix of the Jacobian?

Bottom line: I really think equations 17-27 could be presented in a manner that is less confusing and introduces fewer throw-away symbols.

Lines 196-198: The sentence beginning "Considering ..." is not a sentence.

Line 262: I am confused about whether this is a valid test of the acceleration available from this algorithm, if the global minimum hyperplane first has to be calculated by MAGEmin (i.e., using the old, slow algorithm!).

Figure 2 caption: "Least", not "Levast"; "Conjugate Gradient", not "conjugated gradiend"; "method presented in", not "present in"; "number of minimizations", not "minimization".

Line 299: What is  $\Gamma$ ?

Line 310: The conclusion section starts here, give it a section header.

---

## Author Response (AR1)

We thank the reviewer for reading our manuscript and giving helpful comments that will allow us to clarify our work.

The authors present a modification of their code MAGEMIN to allow for more efficient Gibbs free energy minimization. They state that the advantages of this modification arise from the new minimization algorithm being "unconstrained" as opposed to the "constrained" algorithm used previously. This distinction is misleading and the authors should change the paper to more clearly state the change in strategy. Simply put, the new algorithm is NOT "unconstrained".

We agree with the reviewer that our reformulation is not strictly unconstrained but bound-constrained with the bound being that parameters should be positive. We updated the title of the manuscript accordingly ("A bound-constrained formulation for complex solution phase minimization") and clarified the terminology of the inequality constraints to non-linear inequality constraints.

The central problem is simply stated: minimize the Gibbs free energy of a phase as a function of the fractions of cations $X_i$, I=1,n on one or more crystallographic sites. The minimization is subject to two constraints:
1) The cation fractions must sum to 1:
$sum_i^n X_i = 1$
and
2) they must be non-negative
$X_i >= 0$
Constraint (1) can be implemented either by performing a constrained minimization for which (1) forms an auxiliary statement to the minimization problem OR by minimizing over the null space of the constraint (1). The latter approach is appealing because it reduces the dimensionality of the problem (by the number of crystallographic sites), and avoids the auxiliary statement of constraint.

We agree with the general description of the problem. However, the solution phase formulation as described by Holland et al. (2018) and all previous publications of the THERMOCALC group are given in terms of compositional variables, which within bounds, can results in site fraction < 0.0. There is thus a need to reformulate the solution phase using mixing site constraints. As discussed by the reviewer below, for other (simpler) thermodynamic databases, this may not be necessary.

The primary focus of this paper is on the implementation of the null space approach. And it is on the basis of the removal of the auxiliary statement (1) that they describe their modified algorithm as "unconstrained".

The primary focus of the paper, is to reformulate the compositional and order variable formulation used by THERMOCALC a-x models and subsequently, as moted by the reviewer, remove the auxiliary statement.

However, it is NOT unconstrained. The reason is constraint (2). This must still be dealt with: the minimization must still be subject to the bounds described by constraint 2. Thus the minimization is still constrained, not unconstrained.

We agree that the problem is still bound-constrained and corrected this throughout the manuscript.

In other words, they have removed an equality constraint, but NOT the inequality constraint.

We appreciate the opportunity to clarify our intent. Our intention was not to suggest the removal of the bound constraints of the minimization problem.

Our contribution is that we reformulate the non-linear inequality constraints of the site fraction, which are expressed as functions of compositional and order variables (e.g., Holland et al., 2018), into equality constraints expressed as functions of site fraction variables. Subsequently, we eliminate these equality constraints using a nullspace approach, leaving only the bound constraints on the site fraction variables.

This approach is adopted because the site fraction formulation presented in Holland et al. (2018) incorporates non-linear inequality constraints that restrict the hypercube defined by the 'compositional variables.' In other words, the hypercube defined by the bound constraints of the compositional variables does not fully represent the feasible domain. For instance, certain combinations of compositional variables can result in negative site fractions (even within bounds), thus necessitating the use of these additional inequality constraints. The focus of the paper is the elimination of these inequality constraints.

From the reviewer remarks we understand that this was insufficiently clear in the text, so we will update the manuscript accordingly in the revised version.

Of course the authors DO end up applying the inequality constraint, although this application is somewhat buried in the details (Eq. 32).

Each dimension of the problem is indeed bounded, but as described above, we no longer must invoke a solver with (internal) inequality constraints.

My argument is not with the method itself, nor primarily with the claims of greater efficiency (although see below). It is with what I think is a misleading description of the modification. I implore them to characterize their modification more clearly and correctly. Some further comments.

*We hope that the revised version clarifies this better.*

1) Readers may get the mistaken impression that the null space approach is new in the context of petrological Gibbs free energy minimization codes. It is not, and an example of another minimization code that uses the null space approach is HeFESTo (Stixrude and Lithgow-Bertelloni, 2011). Nor do the authors of HeFESTo claim priority for the null space approach, but it is the closest implementation known to this reviewer in terms of intended application (i.e. to petrology).

*We agree with the reviewer that the nullspace approach is not a novel concept in the context of petrological Gibbs free energy minimization and we did not intent to present it as such. Indeed, this method forms the foundation of the solution phase formalism in THERMOCALC, particularly in the formulation of "compositional variables", which have been used since at least White et al. (2000). These variables are used to parametrize mixing site charge neutrality, enforce the condition that the sum of site fractions equals 1, and ensure that the sum of endmember proportions also equals 1.*

2) The claimed superiority of their new implementation to SLSQP is puzzling to this reviewer. One reason for the source of puzzlement is that HeFESTo uses SLSQP and finds a very high (essentially perfect) rate of success and precision as documented in Stixrude and Lithgow-Bertelloni (2021).

*This is likely because the problem solved using SLSQP in HeFESTo is only bound-constrained, while the one solved in MAGEMin also includes internal inequality constraints. This is why we wanted to reformulate the problem in a similar manner as in Stixrude and Lithgow-Bertelloni (2021).*

Perhaps the reason is that HeFESTo uses the null space approach for the equality constraints and relies on the constraint facility of SLSQP only for the inequality constraint. Maybe in the current paper, the authors are instead relying on SLSQP to take care of the equality AND inequality constraints.

*We think that there is a misunderstanding coming from having a different solution phase formalism in mind here.*
*While the solution phase formalism described in Stixrude and Lithgow-Bertelloni (e.g., 2021, as well as earlier references) is indeed constrained solely by bounds (referred to by the reviewer as inequality constraints), the site fraction formulation presented in Holland et al. (2018) is natively different. It incorporates non-linear inequality constraints that internally restrict the hypercube defined by the compositional variables bounds. For example, certain combinations of compositional variables (within bounds) can still result in a negative site fraction. Therefore, additional non-linear inequality constraints are required. Our manuscript shows how to take those into account before passing the problem to the innermost solver.*

Therefore, to avoid confusion in the literature, I suggest the following test:
Perform the SLSQP minimization(s) again, but by combining SLSQP with the null space approach.

The results of the SLSQP algorithm presented in the manuscript already used the nullspace approach as the solution phase formalism presented in Holland et al., 2018 ensures that the mixing site charge neutrality, the sum (site fraction) = 1 and the sum of endmember proportion = 1 by parameterizing the solution space.

Moreover, using the SLSQP approach on bound-constrained only problems reduces to using a bounded BFGS approach.

3) A final comment on SLSQP. The authors state that SLSQP sometimes fails because it violates inequality constraints and then cannot return to the feasible space. But this should not be true. According to NLOPT documentation SLSQP is guaranteed to respect inequality constraints at all intermediate steps of the minimization.

We respectfully think that the reviewer is mixing bound-constraints and (non-)linear inequality constraints. While bound constraints are indeed always respected, this is not necessarily true for non-linear inequality constraints.

The NLopt documentation indicates the following (https://nlopt.readthedocs.io/en/latest/NLopt_Tutorial/):

*"In principle, we don't need the bound constraint $x_2 \geq 0$, since the nonlinear constraints already imply a positive-$x_2$ feasible region. However, NLopt doesn't guarantee that, on the way to finding the optimum, it won't violate the nonlinear constraints at some intermediate steps, while it does guarantee that all intermediate steps will satisfy the bound constraints. So, we will explicitly impose $x_2 \geq 0$ in order to ensure that the $\sqrt{x_2}$ in our objective is real."*

So, in other words, the nonlinear constraints are not always guaranteed to be satisfied.

Perhaps the problem is that SLSQP sometimes does venture into the space where one or more $X_i$ are exactly zero. If this is the case, it is a problem easily solved, by setting the inequality constraint instead to:
$X_i \geq$ epsilon
very much like their own implementation of the inequality constraint.

As pointed out by the reviewer, having a bound-constraint for a site fraction exactly equal to zero would pose a problem (as log(0.0) is undefined for the configurational entropy term) and we indeed added a small epsilon in the code to avoid this (see https://github.com/ComputationalThermodynamics/SandBox/blob/main/GradientBasedMinimizers/unconstrained_CG_BFGS/functions/gradient_method.jl L99 and L237-245).

We further clarified this in the text LX

REVIEWER #2

This is plainly a useful and important contribution. The essential motivation and results are clear. Nonetheless, before publication some minor revisions are needed to make the presentation as clear and unambigious as it can be.

Lines 40-48: People are starting to explore a new cheap work-around, namely neural net emulators of thermodynamic models. See, for example, https://agu.confex.com/agu/agu24/meetingapp.cgi/Paper/1708197 ... just an abstract at this point, but maybe worth mentioning.

We added the reference, and a couple more on the ongoing development of neural networks

Line 70: Fix grammar: "...inequality constraint in gradient-based methods results in..."

Fixed

Line 77: "...avoids the need to express ..."

Fixed

Line 79: Just "conjugate", not "conjugated"

Fixed

Line 88: "combinations", plural

Fixed

Equation 2: $a_{i(\lambda)}^{id}$ as defined here is NOT an "ideal activity coefficient". It is an "ideal activity". I think everybody agrees that the activity coefficient defines the deviation from ideality and is given by $\gamma_{i(\lambda)}^{id}$ such that $g_{i(\lambda)}^{ex} = RT\log(\gamma_{i(\lambda)}^{id})$. A "coefficient" is a multiplicative quantity that modifies some variable term in an equation. So, for example in the polynomial f(x) = a + bx + cx², the coefficients are a, b, and c. The term "bx" is not a coefficient. Also, better specify, because the usage is not universal, that here "log" means natural logarithm.

We agree and corrected to "activity" term and change log to "ln".

Line 113: "functions"

Corrected

Line 138: c is defined incorrectly. Obviously, it is equal to $X_{Ca}^{M2}$, not $X_{Fe}^{M1}$!

Thank you for spotting it. Fixed accordingly.

Equation 17: The meaning of the Jacobian matrix is vague unless you explicitly spell out the order of the terms in the vectors, i.e. $\boldsymbol{X}_{e_s}^s = \{X_{Mg}^{M1}, X_{Fe}^{M1}, X_{Mg}^{M2}, X_{Fe}^{M2}, X_{Ca}^{M2}\}$ and $\boldsymbol{x}_{cv} = \{x, c, Q\}$.

We agree and updated the manuscript accordingly.

Equation 21: Again, make up your mind: x or xcv?

Fixed to Xcv

Equation 22: What is b? Here it appears to be {1, 1}, but this is never stated. Equation 18 is not the set of equality constraints, it is just the definition of the matrix A (which, again, ought to have a distinctive symbol to indicate it is a matrix).

We clarified this part, by introducing the equality constraint equations Ax = b, introducing the constraint vector b and properly addressing the definition of coefficient matrix A. We also now distinguish the symbols for matrices and vectors by making them bold.

Line 183: It is not obvious (to those that don't do QR decompositions for a living) that Q is a matrix whose columns are q1 ... qm, or that these are ordered into Q1 and Q2 where Q1 has n columns corresponding to the image and Q2 has m–n columns corresponding to the nullspace. Please say that.

We improved this part accordingly

Do you really need to introduce the matrix F? Isn't it just Q2? Or for that matter, Nz as the nullspace matrix of A when A is already the nullspace matrix of the Jacobian?

We corrected this part and use Nz for all instances.

Bottom line: I really think equations 17-27 could be presented in a manner that is less confusing and introduces fewer throw-away symbols.

We improved the presentation and reduced the number of symbols

Lines 196-198: The sentence beginning "Considering …" is not a sentence.

Corrected

Line 262: I am confused about whether this is a valid test of the acceleration available from this algorithm, if the global minimum hyperplane first has to be calculated by MAGEmin (i.e., using the old, slow algorithm!).

The objective here is to evaluate how much a change in the orientation of the Gibbs hyperplane affects the minimization of the solution phases with respect to it. Importantly, this assessment does not require the Gibbs hyperplane to correspond to a global minimum; it can be performed for any arbitrary Gibbs hyperplane, defined by a given set of oxide chemical potentials.
MAGEMin operates in a manner similar to Theriak. First, an initial guess for the Gibbs hyperplane is made (step 1). Next, the solution phase models are minimized with respect to this hyperplane (step 2). The resulting minimized solutions are then used to update the orientation of the Gibbs hyperplane (step 3). Steps 2 and 3 are iterated until convergence, i.e., until the Gibbs hyperplane ceases to change. Between successive iterations, the change in the hyperplane's orientation can be quantified using $\Delta\Gamma$. In this example, we evaluate the performance of the new optimizer with respect to the of magnitude of $\Delta\Gamma$.

We find that the inequality constraint optimizer does not depend on this magnitude while the new bound-constrained optimizer does.

Figure 2 caption: "Least", not "Levast"; "Conjugate Gradient", not "conjugated gradiend"; "method presented in", not "present in"; "number of minimizations", not "minimization".

Corrected

Line 299: What is G?

We added the definition of Gamma: chemical potential of the pure components of the systems (oxides)

Line 310: The conclusion section starts here, give it a section header

We added the header

---

## Referee Report (RR1)

In this work, the authors introduced, in the software MAGEMin (Riel et al., 2022), a more efficient formulation to compute the Gibbs free energy minimization based on the thermodynamic database of Holland et al., (2018). The previous formulation implemented in MAGEMin solved the minimization problem by imposing inequality constraints to bound the fraction of elements in a given crystallographic site between 0 and 1. This formulation is computationally expensive, because it has to be repeated for all the mixing elements in each crystallographic site of the solid solution, thus taking 75-90 % of the total computation time. Moreover, the minimization can occasionally fail due to violations of the inequality constraints, leading to a poor success rate of 57-97 %. The low performances of the inequality constraints formulation, hinders the coupling of phase equilibria calculation with geodynamic models (e.g., self-consistent reactive fluid transport models). The new formulation introduced in MAGEMin by the authors employs the nullspace approach (Ax = 0) to transform the non-linear inequality constraints into a set of linear equalities, which are subsequently turned into a bound-constrained optimization problem: min f(x), with x > 0. This formulation can be solved using gradient-based algorithms, which are computationally efficient. While the nullspace approach is not novel, the authors have further compared the performances of four Gibbs free energy minimization algorithms: two with inequality constraints (CCSAQ and SLSQP), and two with bound-constrained optimization methods (CG and BFGS). Here the authors show that the bound-constrained formulation combined with the BFGS algorithm yields the best performance by significantly reducing the minimization time regardless of the dimensionality of the phase model (i.e. the number of oxides used to define the solid solution), while also maintaining a 100% success rate.

This work remarks the efficiency of the bound-constrained optimization method in computing Gibbs free energy minimization compared to inequality constraints formulations. The algorithm described in this paper represents a valuable computational tool, which can be implemented in other phase equilibria calculators. I believe that the bound-constrained formulation is relevant for the broad geodynamic community, as it will allow more efficient coupling between petrological and thermomechanical models. The quality of the manuscript is good, but some minor revisions are needed before its publication.

**Line 19:** The acronym 'BFGS' in the abstract should be explained before its first use.

**Line 21:** In the conclusions (line 330) you report that the new approach improves the computational time by a factor of ≥ 3; here in the abstract, instead, you report a factor of ≥ 5. Which one is the correct factor?

**Lines 28-30:** there is an extra bracket between 'MELTS and Connolly'.

**Line 37:** the notation 10s 100s is misleading, as it can be read as 10 or 100 seconds. It is better to use the order of magnitude, i.e. $10^1$, $10^2$ milliseconds.

**Line 39:** same as before line 37; it is better to use the order of magnitude.

**Line 59:** The acronym 'CCSAQ' should be explained before its first use.

**Line 62:** GeoPS starts with a capital G. Moreover, it is better to start the sentence introducing what GeoPS is, e.g., "The phase equilibria calculator GeoPS...".

**Line 76:** does *xeos* mean 'compositional equation of state'? The definition of this term is not intuitive, and it should be explicitly defined in the text.

**Line 83:** here the symbol and the unit of Gibbs energy is missing, i.e. $G_\lambda$ and [J].

**Line 85:** does $p_{i(\lambda)}$ represents the mole fraction [mol]? Otherwise, the unit of measurements of Gibbs energy [J] in equation (1) is not consistent.

**Line 86:** the unit of measurements of molar chemical potential $\mu_{i(\lambda)}$ [J mol$^{-1}$] is missing.

**Line 90:** "the *molar* chemical potential of a phase is..." (here molar is missing).

**Line 94-95:** the unit of measurements of the ideal activity $a^{id}_{i(\lambda)}$, the reference Gibbs energy $g^0_{i(\lambda)}$ and the excess energy $g^{ex}_{i(\lambda)}$ are missing. It should explicitly mention that $a^{id}_{i(\lambda)}$ is dimensionless [/]. To satisfy the dimensional consistency of the equation (2), the two energies $g^0_{i(\lambda)}$ and $g^{ex}_{i(\lambda)}$ should be [J mol$^{-1}$]. If this is the case, they should be named: reference molar Gibbs energy and excess molar energy.

**Equation 4, line 101**: why is $N_\lambda$ now referred to olivine ($N_{ol}$)? This has not been established in the text. $N_{ol}$ is also presented in the summation in line 102. The parameters $\phi'_n$ and $\phi_n$ are not described, whereas $\phi_i$ does not appear in the equation.

**Line 102**: can you explain what the asymmetry parameter $v_i$ is?

**Line 104:** the unit of measurements of interaction energy $W_{m,n}$ is missing. It should be [J mol$^{-1}$].

**Equation 5, line 108**: there is too much distance between the introduction of the variable $x_{cv}$ (line 114) and its usage in equation 5 (line 108). What does the subscript _cv stand for? composition (c) and order (v)? This parameter should be introduced before equation 5, e.g.: "Given this formulation, the set of equations 1 to 4 can be directly transformed into the following Gibbs free energy minimization problem as a function of the compositional (c) and order (v) variables $x_{cv}$".

**Line 109:** here you should explicitly introduce the first inequality constrain, i.e. the fraction of the element X $\geq$ 0.

**Line 111:** here you should explicitly introduce the second inequality constrain, i.e. the compositional and order variables must be within an upper (ub$_{cv}$) and a lower (lb$_{cv}$) limit.

**Paragraph 118-126:** Here the authors should make clear to the reader that they are implementing this new formulation in the code MAGEMin based on the thermodynamic database of Holland et al., (2018). Moreover, the authors should report some literature to remark that the introduction of the nullspace approach is not a novel and it has already been used to computed the Gibbs free energy minimization in other phase equilibria calculators (e.g., HeFESTo Stixrude and Lithgow-Bertelloni, 2011 - https://doi.org/10.1111/j.1365-246X.2010.04890.x).

**Line 120:** It is better to avoid having nested brackets e.g., "... crystallographic sites i.e., Bragg–Williams-type formulation (Myhill and Connolly, 2021)".

**Lines 122-123:** too many consecutive adverbs (numerically, significantly, costly). Try with: "which has a significantly higher numerical cost compared to the bound-constrained minimization algorithms".

**Line 131:** the sentence in the bracket can be moved after the equations e.g., "Note that we have dropped the ion charges in the notation of the equations".

**Equation 17, line 155:** It would be clearer to the reader to define x = $X_{es}$ already here.

**Equation 24, line 184:** Is the matrix $N_z$ related to the number of endmembers $N_\lambda$? If yes, could you state it explicitly in the text? If not, wouldn't be better to use a different letter?

**Line 222:** the term '*unconstrained*' should be substituted with '*bound-constrained*'.

**Line 227:** as stated above by the authors (line 220), the case $d_k = -g_k$ has not been explored in this study. Perhaps it is better to remarks this to the reader also in line 227 to avoid confusion. E.g., "... if the iteration increment k = 0 (not investigated here), or..".

**Line 243:** the term '*unconstrained*' should be substituted with '*bound-constrained*'.

**Line 252:** the equation number should in within brackets, e.g., equation (42).

**Line 262:** the compositions (NCKFMASHTO, KNCFMASTOCr, and FMATOCr) should be explained here, since they refer to the oxides components and it might not be intuitive: N = Na2O; C = CaO; etc. Moreover, there should be an explicit reference to Table 1.

**Line 271:** the term '*unconstrained*' should be substituted with '*bound-constrained*'.

**Line 273:** The acronyms 'SLSQP' and 'CCSAQ' should be explained before their first use.

**Table 1:** Temperature should be expressed in [K] not [°C] to be consistent with eq. 2. However, if the authors prefer to keep the units of measurements typically used in metamorphic petrology (kbar and °C) they should state it in the caption of Table 1.

**Figure 2:** It would be useful to report the dimensionality below the mineral e.g., clino-amphibole (dimensionality 10) or (10 oxides composition).

**Line 281:** the term '*unconstrained*' should be substituted with '*bound-constrained*'.

**Figure 3:** It would be useful to report the dimensionality also in this Figure.

**Lines 292, 293, 300:** the terms '*unconstrained*' should be substituted with '*bound-constrained*'.

**Figure 4.** How do you compute the local minimum of the solvus (yellow dots)? Have you obtained them with THERMOCALC using the Holland et al. (2018) database? The reference of these data points must be added.

**Line 304:** when referring to figures and tables you should be consistent throughout the manuscript: either Fig. or figure; Figure or figure; Table or table, etc...

**Line 313:** Figure 5A to 5C.

**Line 315:** 5A, 5B, 5C.

**Line 316:** 5D, 5E, 5F.

**Figure 5:** It would be useful to report the dimensionality also in this Figure.

**Lines 319, 325, 328:** the terms '*unconstrained*' should be substituted with '*bound-constrained*'

**Line 330:** could you elaborate the $\geq 3$ factor improvement? Why is it different from the one reported in the abstract?

**Competing Interests:** This statement should be more explicit e.g., 'BK (co-author) is a member of the editorial board of GMD'.

---

## Author Response (AR2)

In this work, the authors introduced, in the software MAGEMin (Riel et al., 2022), a more efficient formulation to compute the Gibbs free energy minimization based on the thermodynamic database of Holland et al., (2018). The previous formulation implemented in MAGEMin solved the minimization problem by imposing inequality constraints to bound the fraction of elements in a given crystallographic site between 0 and 1. This formulation is computationally expensive, because it has to be repeated for all the mixing elements in each crystallographic site of the solid solution, thus taking 75-90 % of the total computation time. Moreover, the minimization can occasionally fail due to violations of the inequality constraints, leading to a poor success rate of 57-97 %. The low performances of the inequality constraints formulation, hinders the coupling of phase equilibria calculation with geodynamic models (e.g., self-consistent reactive fluid transport models). The new formulation introduced in MAGEMin by the authors employs the nullspace approach (Ax = 0) to transform the non-linear inequality constraints into a set of linear equalities, which are subsequently turned into a bound-constrained optimization problem: min f(x), with x > 0. This formulation can be solved using gradient-based algorithms, which are computationally efficient. While the nullspace approach is not novel, the authors have further compared the performances of four Gibbs free energy minimization algorithms: two with inequality constraints (CCSAQ and SLSQP), and two with bound-constrained optimization methods (CG and BFGS). Here the authors show that the bound-constrained formulation combined with the BFGS algorithm yields the best performance by significantly reducing the minimization time regardless of the dimensionality of the phase model (i.e. the number of oxides used to define the solid solution), while also maintaining a 100% success rate.

This work remarks the efficiency of the bound-constrained optimization method in computing Gibbs free energy minimization compared to inequality constraints formulations. The algorithm described in this paper represents a valuable computational tool, which can be implemented in other phase equilibria calculators. I believe that the bound-constrained formulation is relevant for the broad geodynamic community, as it will allow more efficient coupling between petrological and thermomechanical models. The quality of the manuscript is good, but some minor revisions are needed before its publication.

We thank the reviewer for the very careful read and helpful correction/suggestions that largely improved the quality of the manuscript.

**Line 19:** The acronym 'BFGS' in the abstract should be explained before its first use.

Corrected

**Line 21:** In the conclusions (line 330) you report that the new approach improves the computational time by a factor of ≥ 3; here in the abstract, instead, you report a factor of ≥ 5. Which one is the correct factor?

Thank you for spotting this discrepancy. We corrected the conclusion to ≥ 5.

**Lines 28-30:** there is an extra bracket between 'MELTS and Connolly'.
We added a bracket to close the list of code before the citations.

**Line 37:** the notation 10s 100s is misleading, as it can be read as 10 or 100 seconds. It is better to use the order of magnitude, i.e. 101, 102 milliseconds.

We corrected to "the order of 10 to 100 ms"

**Line 39:** same as before line 37; it is better to use the order of magnitude.

We corrected to Constrained Optimization by Quadratic Approximations (CCSAQ) "from thousands to hundreds of thousands"

**Line 59:** The acronym 'CCSAQ' should be explained before its first use.

We clarified the acronym: "Constrained Optimization by Quadratic Approximations (CCSAQ)"

**Line 62:** GeoPS starts with a capital G. Moreover, it is better to start the sentence introducing what GeoPS is, e.g., "The phase equilibria calculator GeoPS…".

Corrected accordingly

**Line 76:** does *xeos* mean 'compositional equation of state'? The definition of this term is not intuitive, and it should be explicitly defined in the text.

This was corrected as "we present a revised implementation of the compositional and order variables (xeos)" during previous revision. It seems the reviewer did not have accessed to the reviewed manuscript.

**Line 83:** here the symbol and the unit of Gibbs energy is missing, i.e. $G\lambda$ and [J].

We thank the reviewer for spotting this omission. We corrected it accordingly.

**Line 85:** does *pi(λ)* represents the mole fraction [mol]? Otherwise, the unit of measurements of Gibbs energy [J] in equation (1) is not consistent.

We added the unit [mol]

**Line 86:** the unit of measurements of molar chemical potential $\mu i(\lambda)$ [J mol-1] is missing.

Corrected

**Line 90:** "the *molar* chemical potential of a phase is…" (here molar is missing).

Corrected

**Line 94-95:** the unit of measurements of the ideal activity $aidi(\lambda)$, the reference Gibbs energy $g0i(\lambda)$ and the excess energy $gexi(\lambda)$ are missing. It should explicitly mention that $aidi(\lambda)$ is dimensionless [/]. To satisfy the dimensional consistency of the equation (2), the two energies $g0i(\lambda)$ and $gexi(\lambda)$ should be [J mol-1]. If this is the case, they should be named: reference molar Gibbs energy and excess molar energy.

We agree and corrected all points accordingly

**Equation 4, line 101**: why is $N \lambda$ now referred to olivine (*Nol*)? This has not been established in the text. *Nol* is also presented in the summation in line 102. The parameters $\phi'n$ and $\phi n$ are not described, whereas $\phi i$ does not appear in the equation.

We corrected Nol to be N λ to be more generic. Moreover, we clarified equation 4 by correcting th $[\mathrm{J \cdot mol^{-1}}]$ e definition of the terms.

**Line 102**: can you explain what the asymmetry parameter *vi* is?
{m,n}
We added that the asymmetry parameter is the van Laar parameter.

**Line 104:** the unit of measurements of interaction energy *Wm,n* is missing. It should be [J mol-1].

We added the unit

**Equation 5, line 108**: there is too much distance between the introduction of the variable *xcv* (line 114) and its usage in equation 5 (line 108). What does the subscript _cv stand for? composition (c) and order (v)? This parameter should be introduced before equation 5, e.g.: "Given this formulation, the set of equations 1 to 4 can be directly transformed into the following Gibbs free energy minimization problem as a function of the compositional (c) and order (v) variables *xcv*".

We corrected the introduction of the variable x_cv as suggested by the reviewer:
"… the following Gibbs free energy minimization problem as function of the compositional and order variable x_cv"

**Line 109:** here you should explicitly introduce the first inequality constrain, i.e. the fraction of the element $X \geq 0$.

We modified to: "subject to the site fraction of the element"

**Line 111:** here you should explicitly introduce the second inequality constrain, i.e. the compositional and order variables must be within an upper (ubcv) and a lower (lbcv) limit. We corrected this part of the text accordingly

**Paragraph 118-126:** Here the authors should make clear to the reader that they are implementing this new formulation in the code MAGEMin based on the thermodynamic database of Holland et al., (2018). Moreover, the authors should report some literature to remark that the introduction of the nullspace approach is not a novel and it has already been used to computed the Gibbs free energy minimization in other phase equilibria calculators (e.g., HeFESTo Stixrude and Lithgow-Bertelloni, 2011 - https://doi.org/10.1111/j.1365-246X.2010.04890.x).

We agree and added the reference to HeFESTo.

**Line 120:** It is better to avoid having nested brackets e.g., "... crystallographic sites i.e., Bragg–Williams-type formulation (Myhill and Connolly, 2021)".

We removed the nested bracket.

**Lines 122-123:** too many consecutive adverbs (numerically, significantly, costly). Try with: "which has a significantly higher numerical cost compared to the bound-constrained minimization algorithms".

We thank the reviewer for the suggestion and changed the text accordingly

**Line 131:** the sentence in the bracket can be moved after the equations e.g., "Note that we have dropped the ion charges in the notation of the equations".

Corrected

**Equation 17, line 155:** It would be clearer to the reader to define $x = X_{es}$ already here.

We clarify this part by now stating that $x$ and $c$ are the compositional and order variable of olivine as defined in Holland et al., (2018).

**Equation 24, line 184:** Is the matrix $Nz$ related to the number of endmembers $N\lambda$? If yes, could you state it explicitly in the text? If not, wouldn't be better to use a different letter?

Nz is the matrix that span the nullspace of A and its size is function of the number of site fractions and number of equality constraints

**Line 222:** the term '*unconstrained*' should be substituted with '*bound-constrained*'.

Corrected

**Line 227:** as stated above by the authors (line 220), the case *dk = -gk* has not been explored in this study. Perhaps it is better to remarks this to the reader also in line 227 to avoid confusion. E.g., "… if the iteration increment k = 0 (not investigated here), or..".

The conjugate gradient method is initialized for the first iteration as a steepest gradient method and then updated in subsequent iterations of the algorithm. We clarify the sentences to avoid misunderstandings.

**Line 243:** the term '*unconstrained*' should be substituted with '*bound-constrained*'.

Corrected

**Line 252:** the equation number should in within brackets, e.g., equation (42).

Corrected

**Line 262:** the compositions (NCKFMASHTO, KNCFMASTOCr, and FMATOCr) should be explained here, since they refer to the oxides components and it might not be intuitive: N = Na2O; C = CaO; etc. Moreover, there should be an explicit reference to Table 1.

We added the reference to table 1 and now present the chemical system with oxide list.

**Line 271:** the term '*unconstrained*' should be substituted with '*bound-constrained*'.

Corrected

**Line 273:** The acronyms 'SLSQP' and 'CCSAQ' should be explained before their first use.

The acronyms are now explained.

**Table 1:** Temperature should be expressed in [K] not [°C] to be consistent with eq. 2. However, if the authors prefer to keep the units of measurements typically used in metamorphic petrology (kbar and °C) they should state it in the caption of Table 1.

We corrected the temperature unit to K

**Figure 2:** It would be useful to report the dimensionality below the mineral e.g., clino-amphibole (dimensionality 10) or (10 oxides composition).

Corrected

**Line 281:** the term '*unconstrained*' should be substituted with '*bound-constrained*'.

Corrected

**Figure 3:** It would be useful to report the dimensionality also in this Figure.

Corrected

**Lines 292, 293, 300:** the terms '*unconstrained*' should be substituted with '*bound-constrained*'.

Corrected

**Figure 4.** How do you compute the local minimum of the solvus (yellow dots)? Have you obtained them with THERMOCALC using the Holland et al. (2018) database? The reference of these data points must be added.

The spinel solvus has been computed using the Gibbs hyperplane provided in table 1 which was computed using MAGEMin. We added this clarification to the figure 4 caption.

**Line 304:** when referring to figures and tables you should be consistent throughout the manuscript: either Fig. or figure; Figure or figure; Table or table, etc…

Corrected to "Figure"

**Line 313:** Figure 5A to 5C.

We corrected the figure call to Figure 5A-C

**Line 315:** 5A, 5B, 5C.

We corrected the figure call to Figure 5A-C

**Line 316:** 5D, 5E, 5F.

We corrected the figure call to Figure 5D-F

**Figure 5:** It would be useful to report the dimensionality also in this Figure.

Added

**Lines 319, 325, 328:** the terms '*unconstrained*' should be substituted with '*bound-constrained*'

Corrected

**Line 330:** could you elaborate the ≥ 3 factor improvement? Why is it different from the one reported in the abstract?
The first ≥ 3 factor improvement estimate was achieved without considering the minimization failure rate of the CCSAQ and SLSQP methods. When considering it, we find that a ≥ 5 factor improvement to be a better estimate. We corrected this in the text.

**Competing Interests:** This statement should be more explicit e.g., 'BK (co-author) is a member of the editorial board of GMD'.

Corrected